# Pilot Study on the Prevalence of *Entamoeba gingivalis* in Austria—Detection of a New Genetic Variant

**DOI:** 10.3390/microorganisms11051094

**Published:** 2023-04-22

**Authors:** Martina Köhsler, Edwin Kniha, Angelika Wagner, Julia Walochnik

**Affiliations:** Institute of Specific Prophylaxis und Tropical Medicine, Center for Pathophysiology, Infectiology and Immunology, Medical University of Vienna, A-1090 Vienna, Austria

**Keywords:** *Entamoeba gingivalis*, PCR, subtype, gingivitis, periodontitis

## Abstract

*Entamoeba gingivalis* is a parasitic protist that resides in the oral cavity. Although *E. gingivalis* has been frequently detected in individuals with periodontitis, its precise role in this context remains to be established, since *E. gingivalis* is also regularly found in healthy individuals. Sequence data on *E. gingivalis* are still scarce, with only a limited number of sequences available in public databases. In this study, a diagnostic PCR protocol was established in order to obtain a first impression on the prevalence of *E. gingivalis* in Austria and enable a differentiation of isolates by targeting the variable internal transcribed spacer regions. In total, 59 voluntary participants were screened for *E. gingivalis* and almost 50% of the participants were positive, with a significantly higher prevalence of participants with self-reported gingivitis. Moreover, in addition to the established subtypes ST1 and ST2, a potentially new subtype was found, designated ST3. 18S DNA sequencing and phylogenetic analyses clearly supported a separate position of ST3. Interestingly, subtype-specific PCRs revealed that, in contrast to ST2, ST3 only occurred in association with ST1. ST2 and ST1/ST3 were more often associated with gingivitis; however, more data will be necessary to corroborate this observation.

## 1. Introduction

*E. gingivalis* is an amoeboid protist inhabiting the oral cavity of humans, but has also been detected ectopically, e.g., in a pulmonary abscess, neck nodule or osteomyelitis of the mandible [1], and just recently in the nasal cavity of a greyhound [2]. *E. gingivalis* does not produce cysts and is believed to be transmitted directly from human host to human host, as no other reservoir has been clearly established yet.

There are still controversies about the pathogenic potential of these amoebae. With the rather recent establishment of molecular techniques for detection, several studies have found evidence for a clear correlation between *E. gingivalis* infection and periodontal disease [3,4,5]. However, irrevocable proof of its involvement in establishing periodontitis is still lacking, and the fact that quite a number of healthy individuals also test positive for *E. gingivalis* hampers a definite conclusion [1,6]. Nevertheless, *E. gingivalis* seems to be able to invade the oral mucosa and damage host cells via a mechanism called trogocytosis [5,6]. Amoebic trogocyosis was first described for the related *Entamoeba histolytica*, the causative agent of amoebic dysentery [7]. By employing trogocytosis, *E. histolytica* can actively take over host membrane proteins upon contact with host cells and integrate them on its surface. This has been proposed as an evasion mechanism, preventing the lysis of *E. histolytica* by host serum, while ultimately leading to host cell death [8,9].

The pathogenic potential of *E. gingivalis* was further corroborated by the observation that its interaction with gingival cells leads to the expression of TNF, IL8, IL11 and proinflammatory chemokines [10]. Moreover, *E. gingivalis* has been attributed a possible role as an opportunistic pathogen in the oral cavity of immunocompromised patients [11,12]. In summary, although *E. gingivalis* might not act as the sole etiological agent in cases of periodontitis, it also seems to be unlikely that it is only a harmless commensal inhabiting periodontal pockets and not contributing to disease at all. 

Currently, only limited molecular data on these amoebae are available. To date, two *E. gingivalis* subtypes have been established, namely subtype 1 (ST1) and subtype 2 kamaktli (ST2) [13,14]. Apart from being genetically distinct, a different pattern of infectious behavior for these STs was proposed and the question was raised whether these STs represent separate species [15]. The internal transcribed spacers ITS1 and ITS2, located between the 18S rRNA gene and the 28S rRNA gene and separated by the 5.8S rRNA gene, have been shown to be useful tools to differentiate between closely related eukaryotic species or even genotypes within one species [16,17]. The ITSs evolve more quickly, which is reflected in a greater sequence variability compared to ribosomal genes [18]. This was reported for several *Entamoeba* species but was also shown for *E. gingivalis* between ST1 and ST2 [14,19]. 

The main aim of this study was to establish a sensitive PCR for the detection of *E. gingivalis*, additionally enabling a more detailed differentiation of isolates by amplifying and subsequent sequencing of a more variable DNA fragment comprising both internal transcribed spacers. Since no studies on the incidence of *E. gingivalis* in Austria exist to date, another aim of this study was to obtain a first impression of the prevalence of *E. gingivalis* in generally healthy individuals. In order to include some basic information on the subjective health status of the oral cavity of the participants, participants who claimed that they regularly suffer from symptoms such as bleeding, irritation, inflammation and pain in the gingiva were considered as participants with self-reported gingivitis (Self-RGI) in this study. Gingivitis is defined as the earliest stage of periodontitis and by including Self-RGI it was attempted to show that *E. gingivalis* might be more prevalent long before the onset of more severe periodontal diseases. 

## 2. Materials and Methods

### 2.1. Sampling/DNA Extraction

Participants were recruited with flyers asking whether they wanted to find out if they carry parasites in their oral cavity. Prior to sampling, the patients signed an informed consent form. The flyers and the consent form had been approved by the Ethics Committee of the Medical University of Vienna: vote 1228/2022. 

The participants were asked whether they regularly suffer from redness, swelling, bleeding, inflammation or pain in their gingiva. Participants who answered these questions in the affirmative were considered as participants suffering from self-reported gingivitis (Self-RGI) in this study. Sampling took place at our institute at the Department of Molecular Parasitology. Samples were taken by the participants themselves, wearing sterile gloves, from at least three different sites in the oral cavity with sterile interdental brushes, which had been additionally treated with UV light prior to sampling. Interdental brushes were collected in sterile 15 mL tubes with 200 µL sterile saline. Subsequently, tubes were vortexed, and samples were centrifuged for 1 min at 700× *g*. Interdental brushes were removed, and the samples were individually transferred into 1.5 mL reaction tubes for immediate DNA extraction or stored at −20 °C until further processing. DNA was isolated with the QIAamp^®^ DNA Mini Kit 250 (QIAGEN, Hilden, Germany) following the manufacturer’s instructions. In the last step, DNA was resuspended in 50 µL of AE buffer. In addition to the samples of the participants, DNA was also isolated from the corresponding saline without a sample in order to avoid false-positive results due to contamination. 

Our initial sampling approach was applying a commercially available sterile saline solution of pharmaceutical quality in individual vials as a mouthwash, but this method proved unreliable due to the occasional detection of *E. gingivalis* DNA in the saline solution. This might be explained by the high prevalence of *E. gingivalis* in the population, possibly leading to traces of *E. gingivalis* DNA in tap water and thus in preparations based on it. Therefore, in the final setup, all components used for sampling and during DNA isolation were tested for contamination and DNA-free water was used for PCR. 

### 2.2. Primer Design

Specific primers for the detection of *E. gingivalis* were designed based on all *E. gingivalis* DNA sequences available in GenBank and on representative sequences from other *Entamoeba* species. Initially, primers targeting the end of the 18S rRNA gene and the beginning of the 28S rRNA gene were used (EGITS). Subsequently, primers amplifying altogether four overlapping fragments of the 18S rRNA gene of *E. gingivalis* were designed (EGF1-EGF4) in order to obtain complete 18S rRNA gene sequences from a selection of strains representing all known subtypes. In order to detect potential double infections, primers specific for all three subtypes were constructed and run with all positive samples. For ST1 and ST3, two specific primer pairs were designed (ST1 and ST3); for ST2 one specific primer (ST2fw) was combined with EGF2rev. Primers were synthesized by Microsynth (Balgach, Switzerland). All primers used in this study are listed in Table 1. 

### 2.3. PCR/Sequencing

PCRs were run on an Eppendorf Mastercycler (Eppendorf AG, Hamburg, Germany) in reaction volumes of 50 µL, containing 10X reaction buffer B, 2.5 mM MgCl_2_, 1.6 mM dNTPs, 1 µM primers, 1.25 units DNA polymerase and 1.5 µL DNA in DNA-free H_2_O. The gene fragments were amplified using the following conditions: 95 °C for 15 min, followed by 30 cycles of 95 °C for 1 min, 56 °C for 45 s and 72 °C for 1 min, followed by a final extension of 72 °C for 7 min. A sample without DNA and a DNA preparation from the corresponding saline for sampling were run as negative controls. One sample positive for *E. gingivalis* based on microscopical observation was used as a positive control. The obtained DNA fragments were separated by electrophoresis on a 2% agarose gel stained with GelRed and visualized with a Gel Doc™ XR+ Imager (Bio-Rad Laboratories, Inc., Hercules, CA, USA). Bands were cut out of the gel and purified with an Illustra™ GFX™ PCR DNA and Gel Purification kit (GE Healthcare, Buckinghamshire, UK). 

Sanger sequencing was performed with the Applied Biosystems SeqStudio Genetic Analyzer (Thermo Fisher Scientific, Waltham, MA, USA). At least two sequences were obtained from both strands. For all samples, the fragments obtained from the EGITS primers (containing the ITS1, 5.8S rRNA gene and the ITS2) were sequenced. For the samples HA, RS, ZM, IH, RS, DM, ALA, BF and SGL, the entire 18S rRNA gene was additionally sequenced with primers EGF1–EGF4. In order to confirm the specificity of the subtype-specific PCRs, for at least two samples per subtype, PCR products were sequenced for subtype confirmation. 

### 2.4. Sequence Assembly

Consensus sequences were generated using the DNA sequence analysis tool GeneDoc 2.7.0. For strains HA, RS, ZM, IH, RS, DM, ALA, BF and SGL, 18S rRNA gene fragments and EGITS fragments were assembled to full-length sequences. All sequences were deposited in GenBank and are available under the following accession numbers: OP161459-OP161476 (EGITS) and OQ225449-OQ225457 (18S/EGITS). Sequences were aligned with ClustalX 2.1 [20] and refined manually in GeneDoc for better consensus. Similarity matrices were calculated for the 18S rRNA gene, the EGI fragment and the ITS1 and the ITS2 fragments in order to evaluate sequence identities between all investigated strains and sequences available from GenBank.

### 2.5. Phylogeny

Multiple alignments for phylogenetic analyses were performed using Muscle in MEGA software version 11 [21]. Three sequences representing ST1 (KX027297, KX027298 and D28490) and ST2 (KX027294, KX027295 and KX027296) were analyzed together with sequences obtained in this study. Two sequences representing *E. suis* were used as an outgroup (DQ286372 and LC1230019). 

Based on best-fit evolutionary model selection, the Hasegawa–Kishino–Yano model specifying a gamma distribution and invariable sites was applied [22]. The algorithms used for phylogenetic analysis were maximum likelihood (ML), maximum parsimony (MP) and the Neighbor-Joining method (NJ). All three analyses were run with 1.000 bootstrap replicas. In the final dataset, 1817 positions were analyzed for ML and MP and 1939 positions for NJ. 

### 2.6. Statistical Analysis 

To test whether there is a statistically significant correlation between the presence of *E. gingivalis* and gingivitis, two-sided Fisher’s exact test (FET) was employed. Calculations were performed with GraphPad Prism. 7.00. 

## 3. Results

### 3.1. Participants

A total of 59 samples were investigated in the course of this study. The proportion of male and female participants was well balanced. Details on the age, gender and Self-RGI of the participants are given in Table 2.

### 3.2. Detection of Entamoeba gingivalis DNA and Association with Self-RGI

Of the 59 samples investigated, DNA of *E. gingivalis* was detected in 27 study participants using primers EGITSfw and EGITSrev. The results are summarized in Table 3. The prevalence was considerably higher in females (56.7%) than in males (34.5%), albeit not statistically significant. There was no significant difference between the three age groups; however, in the group aged 50 and older, slightly fewer were positive. Of the 59 participants, 18 (30.5%) specified that they suffered from mild to moderate Self-RGI at least occasionally. Of these 18 participants, 14 (77.8%) were positive for *E. gingivalis.* Accordingly, only four (22.2%) of the participants who specified that they suffer from Self-RGI tested negative. All of these participants were in the above-50 age group. On the other hand, 31.7% of the participants who never experienced any gingival discomfort tested positive for *E. gingivialis*. The correlation between Self-RGI and the presence of *E. gingivalis* was shown to be statistically significant with a *p*-value of 0.0016. 

### 3.3. Sequence Analysis 

DNA sequence data were obtained from all positive samples in order to rule out unspecific results on the one hand and for phylogenetic analyses on the other hand. Sequences were compared to published *E. gingivalis* sequences available in GenBank. To date, only four *E. gingivalis* sequences containing the ITS1 and ITS2 region are available, two representing ST1 and two representing ST2. Interestingly, while thirteen samples were highly similar to ST1 (93–100% sequence identity) and eight samples to ST2 (93–98% identity), five samples were dissimilar to both published STs. These sequences, however, were highly similar to each other with 99 to 100% sequence identity, but only 85–86% and 80–81% identity to ST1 and ST2, respectively. Based on these findings, it was assumed that they belong to a yet-undescribed subtype. In this study, this potentially new ST is referred to as ST3. In one sample (CG), the presence of two STs, namely ST1 and ST3, was revealed. 

In order to obtain more sequence information on ST3, specific primers amplifying the entire 18S rRNA gene of *E. gingivalis* were designed. 18S sequencing was performed on three representatives of ST3 (ALA, BF and SGL) and three representatives of both ST1 (HA, DS and ZM) and ST2 (RS, DM and IH) to enable a more detailed comparison of the different STs. Full-length 18S rRNA gene sequencing corroborated the high dissimilarity of ST3 to both ST1 and ST2. Dissimilarities were found throughout the 18S gene, with unique variable regions for all three STs. ST3 and ST1 display a higher degree of sequence identity than either ST1 or ST3 to ST2. Sequence identities between all three STs in the 18S rRNA gene, ITS1, 5.8S rRNA gene, ITS2 and the entire sequence ranging from the beginning of the 18S rRNA gene to the beginning of the 28S rRNA gene are given in Table 4. Additionally, identities in the 18S rRNA gene with *E. suis*, which has been shown to be the most closely related *Entamoeba* species, and with *E. histolytica* and *E. dispar*, are shown to demonstrate the variability of ITS regions. 

As expected, sequence identities between *Entamoeba* strains were higher in the 18S rRNA gene and the 5.8S rRNA gene. For *E. gingivalis*, the ITS1 appears to be the most variable region with less than 70% sequence identity between different subtypes, but even within one ST variabilities are considerably higher than in the ribosomal genes, potentially enabling a subdivision within one ST. Comparing different *Entamoeba* species, however, the ITS2 was more variable, with only 31% to 36% sequence identities with *E. histolytica* and *E. dispar*. 

### 3.4. Detection of Double Infections

Since infections with more than one ST have been reported previously and results for the sample CG indicated two different *E. gingivalis* strains in the sample, all positive samples were additionally tested for potential double infections employing primers specific for all three STs. In Table 5, the final number of positive samples including information on the respective subtype and potential double infection is provided. An image of an agarose gel showing two samples for ST1, ST2 and double infections with ST1/ST3, respectively, and two samples which tested negative with all three primer pairs, is shown in Figure 1. 

In nine samples, double infections were detected, which all were shown to be associated with ST1 and ST3. In fact, in all samples initially positive for ST3, ST1 could be detected as well with the specific primers. In 3 of the 13 samples positive for ST1, DNA of ST3 could also be detected, while in none of the samples positive for ST2 was a second subtype detectable (Table 5). 

Due to the relatively small sample size, of course, only limited conclusions can be drawn about the distribution of STs; however, in our study it was shown that ST1 seems to be the predominant subtype in men, mixed infections with ST1/ST3 appear to be more prevalent in the age group from 35 to 50, while *E. gingivalis* altogether appears to be less frequent in the group over the age of 50. A more detailed analysis of the STs in samples from participants with Self-RGI showed that ST2 and ST1/ST3 were slightly more often associated with Self-RGI, with 63% and 56% of the positive samples, respectively, and ST1 alone was detected the least often in individuals with gingivitis, with 40% of the positive samples. 

### 3.5. Phylogenetic Analysis

In Figure 2, a consensus phylogenetic tree is shown demonstrating the position of ST3 within the species of *E. gingivalis*. *E. suis*, being the most closely related species to *E. gingivalis*, was chosen as an outgroup. Trees produced with all three different algorithms (ML, MP and NJ) all supported a separate position of ST3 with high bootstrap values. With all methods, a closer relationship of ST3 to ST1 was supported, while ST2 unambiguously forms a separate clade within the species.

## 4. Discussion

In this study, we established a highly sensitive PCR protocol for the detection of *E. gingivalis* in samples taken from the oral cavity, targeting the variable ITS1 and ITS2 regions. A high prevalence of *E. gingivalis* was found in individuals with and without Self-RGI. Interestingly, more females than males were affected and there was a statistically significant correlation between *E. gingivalis* detection and Self-RGI. Moreover, a new subtype of *E. gingivalis* was found.

It is generally complicated to compare prevalence data on *E. gingivalis* obtained from different studies, since several aspects have to be considered, including the respective sampling technique, the background of the participants concerning periodontal disease, and the method of detection [23]. While in most studies sampling was performed by trained specialists, in our study the participants took the samples themselves. 

In contrast to other studies, in this study individuals without diagnosed periodontal disease were tested for the presence of *E. gingivalis* DNA in their oral cavity. Based on affirmative answers to questions about bleeding, irritation, redness, inflammation and pain in the gingiva, the participants were subdivided into a “healthy” group and a group with Self-RGI. In other studies, the rate of positivity in the periodontitis cohort was similar to the group with Self-RGI in our study and ranged from approximately 60% to 80%; however, in these studies all samples were taken by trained professionals directly from periodontal pockets, with a defined minimal depth ranging from 3 to 7 mm [3,4,15,24]. Of the study participants without any symptoms, approximately 30% had positive PCR results, which is higher than in several other studies [3,15,24,25]. Bonner et al. (2014) reported similar percentages of positive PCR clinically negative patients, but mentioned that due to low amounts of DNA and PCR inhibition, patients had to be excluded from the final PCR, potentially influencing the result. Generally, PCR has a higher sensitivity compared to microscopy, as demonstrated in several studies [12,25]. 

Of course, the results obtained in this study must be interpreted with caution since Self-RGI is strongly dependent on the awareness, sensitivity and pain threshold of a participant. Additionally, the sampling method we employed is more prone to inconsistencies and not comparable to sampling performed by trained professionals. 

A very interesting finding was that in addition to ST1 and ST2, a third, previously unknown subtype was detected, designated ST3 in this study. The PCR protocol employed was newly established in the course of this study and primers are located in a more conserved DNA region, enabling the detection of more genetic subtypes than reported before. Most studies used PCRs only targeting ST1, partly due to limited sequence data available [4,12,26,27]. After the discovery of ST2, PCRs targeting both STs were employed [14,15,23]. However, sequence comparisons during the current study revealed that all primer combinations employed before would not amplify DNA from ST3, which might explain why this ST has not been detected before. 

In our study, ST1 clearly was the predominant subtype, detected in 19/59 individuals tested, including double infections with ST3. ST2 was found in eight individuals and ST3 in nine individuals, the latter all being co-infected with ST1. It is peculiar that ST3 was never detected without co-infection, but it is too early to conclude that the association of ST3 with ST1 is obligatory, since the number of samples in this study was relatively small. Nevertheless, a symbiosis of these two types cannot be ruled out. Based on sequence similarity and phylogenetic analysis, ST3 and ST1 also appear to be more closely related to each other than they both are to ST2, but a distinct position of ST3 within the species *E. gingivalis* was supported in all analyses with high bootstrap support. The consistent association of ST3 with ST1 also raised the question of whether ST3 might result from intra-strain variation within the SSU rDNA. Although rDNA repeats are generally considered to undergo concerted evolution [28], intra-genomic rDNA polymorphism has been frequently reported [29,30]. Arguments against intra-strain variation are that sequence dissimilarities between ST1 and ST3 are apparent in various variable regions throughout the entire 18S rRNA gene and the ITS1-5.8S-ITS2 portion, and exceed the variabilities reported previously [30,31]. Furthermore, polymorphisms, in particular in the ITS1 region, are typically located in AT-rich portions with varying repeats of the same nucleotide pattern or single point mutations, as reported for *Dientamoeba fragilis* [32]. However, this is not the case for the variation between ST1 and ST3, where differences between the subtypes are generally located in variable regions; similar to differences between subtypes ST1 and ST3 and ST2 and other *Entamoeba* species. These differences usually affect several nucleotides. Additionally, the fact that there is no evidence for intra-strain variation in ITS1-5.8S-ITS2 sequences in *E. histolytica* supports the assumption that the amplified DNA from ST1 and ST3 stems in fact from two distinct strains [19,32]. 

In the original description of ST2 [14], it was proposed that this subtype might represent a new species due to the highly variable ribosomal sequences. Compared to 18S rDNA sequence identities of 98% between *E. histolytica* and *E. dispar*, two well-established distinct species, sequence identities of only 91% between ST1 and ST2 support this theory. However, the description of a new species in the genus *Entamoeba* is complicated. On the one hand, taxonomy is based on morphology, with the number of nuclei in the mature cysts being the defining morphological feature [33]. On the other hand, in the case of morphologically identical species such as *E. histolytica* and *E. dispar*, molecular data eventually provide the evidence for a separation into two distinct species [34]. Stensvold et al. (2011) established so-called identification tags for *Entamoeba* spp. in the absence of morphological data and defined subtypes as well supported phylogenetic clusters within a defined species [35]. Based on this scheme, ST2 as well as ST3 would represent subtypes. To date, there are no clear specifications as to what defines an *Entamoeba* species based on molecular data, and without morphological data, a proposed system might remain artificial and pose the risk of false assignments. In the case of ST3, we propose a new subtype rather than a distinct species, since sequence identities of 95% between ST1 and ST3 are clearly higher than between ST1 and ST2 with 88 to 89% identity in the 18S rRNA gene. Additionally, for *Entamoeba coli*, a third subtype has been proposed just recently with 7% and 13% sequence dissimilarity to previously defined sequence types, but is only based on just a 484 bp portion of the 18S gene [36]. In the current study, the entire 18S rRNA gene was used for phylogenetic analyses. Additionally, the designation of a new subtype within the species *E. gingivalis* is corroborated when sequences of the ITS1 and ITS2 are compared. For all three subtypes, the variability in the ITS regions was clearly higher compared to the 18S gene, possibly enabling a sub-division within one ST, but between different STs these variabilities were even more pronounced, in particular for ITS2 sequences. A monophyletic relationship between ST1, ST2 and ST3 still was supported when ITS sequences were compared to the ITSs of *E. histolytica* and *E. dispar* sequences, with less than 50% identity in both ITS1 and ITS2. Generally, the ITS regions are more variable than 18S rRNA genes and their utility for species differentiation has been demonstrated for several protozoa, such as *Giardia* spp., *Trichomonas* spp. and *Leishmania* spp. [37,38,39]. Additionally, for *E. histolytica* and *E. dispar*, the spacer regions have been shown to be a more suitable tool for differentiation than the 18S rRNA gene [40]. 

Subtype-specific PCRs were established to rule out mixed infections, since Garcia et al. [15] reported a high number of mixed infections and in one of our samples an infection with two STs was indicated. However, since sequencing enabled the unambiguous determination of STs, it was quite surprising that in nine samples both ST1 and ST3 were detectable. For *E. bovis*, a similar situation has been described, where the use of different primers also revealed double infections, not detectable with the initial PCR approach [35]. It remains to be determined why, in the case of mixed infections, only one ST could be detected initially, since mixed DNA in a sample often impairs sequencing by producing poor quality sequences with a low signal-to-noise ratio. It is possible that one ST might be the predominant ST, more prone to be amplified in PCR. Approaches with qPCR in the future might shed light on this issue. 

The authors are aware that the study design is subject to some limitations since this approach was designed as a pilot study with the primary aim of establishing a detection method to enable a more detailed differentiation of *E. gingivalis.* The sampling method is prone to inconsistencies because it was not performed by trained specialists. Self-reported gingivitis strongly depends on the participant’s subjective mindset, and the fact that no additional health information of the participants was collected limits a more conclusive interpretation of the results. 

## 5. Conclusions

A surprisingly high prevalence of *E. gingivalis* was found in our study, which represents the first screening for this organism in Austria. A new PCR protocol was established and a new subtype, designated ST3, was detected, which occurred invariably associated with ST1. It remains to be established whether this relationship is stable and obligatory. Hopefully, future studies with a more sophisticated study design, a higher number of samples, including more participants with clinically diagnosed periodontitis, and detailed information obtained with a well-designed questionnaire will shed more light on the still-unresolved issues concerning the species *E. gingivalis*. 

## Figures and Tables

**Figure 1 microorganisms-11-01094-f001:**
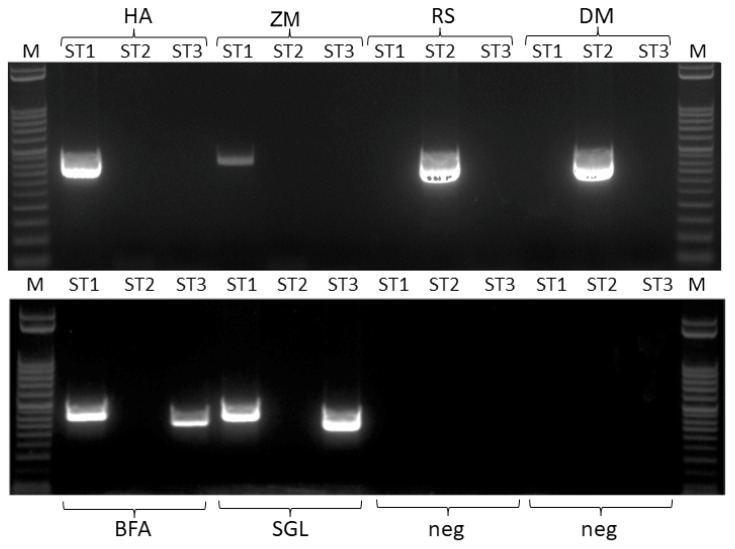
Agarose gel of amplicons generated with sequence-type-specific primers ST1fw/ST1rev, ST2fw/EFF2rev and ST3fw/ST3rev for two samples representative for ST1 (HA and ZM), two samples representative for ST2 (RS and DM), two samples representative for ST1/ST3 double infection (BFA and SGL) and two samples negative for *E. gingivalis* DNA (neg); M, marker lane.

**Figure 2 microorganisms-11-01094-f002:**
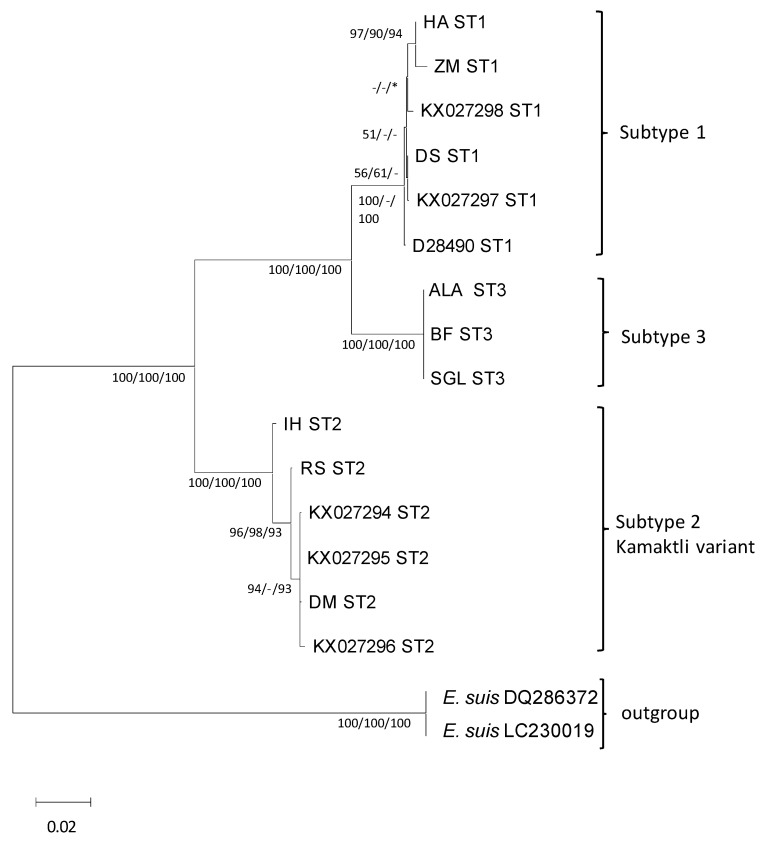
Rooted phylogenetic tree based on 18S rRNA gene sequences using the maximum likelihood method and Hasegawa–Kishino–Yano model [22]. The tree is drawn to scale, with branch lengths measured in the number of substitutions per site. All positions containing gaps and missing data were deleted. There were a total of 1817 positions in the final dataset. Two sequences of *E. suis* were chosen as outgroup. Bootstrap values are based on 1.000 replicates and are given at the nodes (ML, NJ and MP). Hyphens indicate bootstrap values below 50, asterisks indicate missing branches. Bar: 0.02 substitutions per site.

**Table 1 microorganisms-11-01094-t001:** List of primers used in this study.

Primer	Primer Sequence (5′-3′)	Amplicon Length (bp)	Average Tm (°C)
EGF1fw	CTGATGTTAAAGATTAAGCCATGC	≈590	60.3 °C
EGF1rev	CGAGCCTTTTAATCACAACAAC		58.4 °C
EGF2fw	GTTGTTGTGATTAAAAGGCTCG	≈507	58.4 °C
EGF2rev	GAAGTTCATACACTCAAGATTTCTC		60.9 °C
EGF3fw	GAGAAATCTTGAGTGTATGAACTTC	≈470	60.9 °C
EGF3rev	CCAAGATGTCTAAGGGCATCAC		59.5 °C
EGF4fw	GTGATGCCCTTAGACATCTTG G	≈360	59.5 °C
EGF4rev	CTCTAAATAAGGAGGTTCACATC		59.2 °C
EGITSfw	GATGTGAACCTCCTTATTTAGAG	≈500	59.2 °C
EGITSrev	GATATGCTTAAGTTAAGAGAGTCAT		59.2 °C
ST1fw	GGAGTAAAAAGAAACAGTAGTAAG	≈400	57.6 °C
ST1rev	CCAATTACTCACATTACAACAATC		58.3 °C
ST2fw	CTCAACGAAGACAATAGAGAAG	≈420	58.4 °C
EGF2rev	GAAGTTCATACACTCAAGATTTCTC		60.9 °C
ST3fw	CTCTACGTAACTTGTTACAAGAGAGG	≈330	64.6 °C
ST3rev	TAATTATCTCCATTTCTCTTCAAAATG		59.2 °C

**Table 2 microorganisms-11-01094-t002:** Demographic characteristics of participants.

	Total	Self-RGI ^1^	Healthy	Age Range	Age Ø + SD
Total	59	18	41	21–81	44.5 ± 15
Women	30	11	19	21–79	43 ± 15
Men	29	7	22	22–81	46 ± 16

^1^ Self-RGI (self-reported gingivitis).

**Table 3 microorganisms-11-01094-t003:** Results of the *E. gingivalis* PCR according to gender and age groups. Number and percentage of participants suffering from Self-RGI are given in the last three columns.

	Total	Positive	Negative	Self-	Pos./Self-	Neg./Self-	No GI ^1^	Pos./H ^2^	Neg./H ^2^
		No. (%)	No. (%)	RGI ^1^	RGI ^1^ No. (%)	RGI ^1^ No. (%)		No. (%)	No. (%)
Total	59	27 (45.8%)	32 (54.2%)	18	14 (77%)	4 (22%)	41	13 (32%)	28 (68%)
men	29	10 (34.5%)	19 (65.5%)	7	6 (86%)	1 (14%)	22	4 (18%)	18 (82%)
women	30	17 (56.7%)	13 (43.3%)	11	8 (73%)	3 (27%)	19	9 (47%)	10 (53%)
Age up to 35	20	9 (45.0%)	11 (55.0%)	5	5 (100%)	0	15	4 (27%)	11 (73%)
Age 35–50	21	11 (52.4%)	10 (47.6%)	6	6 (100%)	0	15	5 (33%)	10 (67%)
Age from 50	18	7 (38.9%)	11 (61.1%)	7	3 (43%)	4 (57%)	11	4 (36%)	7 (64%)

^1^ Self-RGI (self-reported gingivitis); ^2^ H (healthy).

**Table 4 microorganisms-11-01094-t004:** Sequence identities between *E. gingivalis* sequence types ST1, ST2 and ST3 and *E. histolytica* and *E. dispar* in the ribosomal RNA, including the 18S rRNA gene, internal transcribed spacer 1, 5.8S rRNA gene and internal transcribed spacer 2 and between *E. gingivalis* and *E. suis* in the 18S rRNA gene.

*Entamoeba* ST	Comparison ST	Identity Including Gaps (%)					
		18S	EGI	18S-28S	ITS1	5.8S	ITS2
ST1	ST1	98–99	93–100	97–99	88–100	99–100	90–100
ST1	ST2	88–89	78–81	86–87	61–69	91–92	66–74
ST1	ST3	94–95	85–86	92–93	65–68	97–98	80–86
ST2	ST2	98–99	92–100	96–99	84–100	96–100	87–100
ST2	ST3	88–89	80–81	86–87	64–68	91–94	69–72
ST3	ST3	100	99–100	99–100	99–100	99–100	97–100
ST1–ST3	*E. suis*	84–85	–	–	–	–	–
ST1–ST3	*E. histolytica*	76–77	54–58	71–72	42–52	67–69	31–36
ST1–ST3	*E. dispar*	75–76	*53–58*	70–71	42–53	66–68	31–36

**Table 5 microorganisms-11-01094-t005:** Distribution of STs in positive samples after sequence-type-specific PRCs, according to gender and age groups.

	Pos.	Pos.	Pos.	Pos.	Neg.	Total
	No. (%)	ST1	ST2	ST1/ST3	No. (%)	
Total	27 (46%)	10 (37%)	8 (30%)	9 (33%)	32 (54%)	59
Men	10 (34%)	5 (50%)	2 (20%)	3 (30%)	19 (66%)	29
Women	17 (57%)	5 (30%)	6 (35%)	6 (35%)	13 (43%)	30
Age up to 35	9 (45%)	4 (45%)	3 (33%)	2 (22%)	11 (55%)	20
Age 35–50	11 (52%)	3 (27%)	2 (18%)	6 (55%)	10 (48%)	21
Age from 50	7 (39%)	3 (43%)	3 (43%)	1 (14%)	11 (61%)	18

## Data Availability

Data available on request.

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
