# Peer review of "Pilot Study on the Prevalence of Entamoeba gingivalis in Austria—Detection of a New Genetic Variant"

_microorganisms, 2023, doi:10.3390/microorganisms11051094_

Round 1
Reviewer 1 Report
This article is a valuable observational study on the infection status of Entamoeba gingivalis of which details are still unknown, and the discovery of a new detection method and an unknown subtype is highly valuable. The experimental results obtained are clinically suggestive and may provide fundamental information, especially in the analysis of oral microflora. However, there are some expressions that are difficult for readers to understand, so the following points could be improved to make the research report even better.
Introduction section
・It is not clear which target diseases are of greatest concern as a result of infection with E.gingivalis; whether the research target is gingivitis or periodontitis. If the authors want to investigate microflora in healthy subjects, it is difficult to understand why gingivitis patients are included. The research objectives should be simple, clear and easily understood by the reader.
・The authors examined whether E.gingivalis is a factor in the development of gingivitis, but gingivitis without periodontal tissue destruction is not a very serious disease. Gingivitis and periodontitis are also mixed in the text. A clear stating of the disease to be investigated is necessary.
Material and Methods section
・Basic medical information on the participants is unclear. Specifically, the number of teeth present, oral hygiene status, degree of periodontal disease, dental history, smoking habits, and history of antibiotic use in the past several months. Age and gender alone equals no information.
・P2L84-90
Shouldn't commercial products be contaminated with E.gingivalis.?
・P3L125
The consecutive numbers in the subtitle are wrong. Need to be corrected thereafter.
Discussion section
・P9L267-268
It is unclear why the patients themselves would collect the samples. The method is uncertain and the detailed patient profile is unknown. Are Volunteers Dental Professionals? Patient profiles need to be mentioned in the Methods section.
・P9L269-
If the authors assume that the purpose of the study is to investigate the bacteria that cause periodontal disease, isn't it inconsistent to select a sample of healthy subjects?
・Throughout the article, there is no mention of bias and no mention of limitation.
Author Response
This article is a valuable observational study on the infection status of Entamoeba gingivalis of which details are still unknown, and the discovery of a new detection method and an unknown subtype is highly valuable. The experimental results obtained are clinically suggestive and may provide fundamental information, especially in the analysis of oral microflora. However, there are some expressions that are difficult for readers to understand, so the following points could be improved to make the research report even better.
General reply:
Thank you very much for your valuable and constructive comments. Your remarks are well-founded and gave us a better understanding how our manuscript can be improved.
Please find changes to the manuscript highlighted in yellow.
We hope that we could articulate the actual aim of our study in a better way, which was to establish a detection method for Entamoeba gingivalis (possible enabling a more detailed differentiation) while getting a first idea of the general occurrence of E. gingivalis in Austria. Since the prevalence in healthy individuals is fairly high as reported in other studies, we felt that getting samples from volunteers might be sufficient to proof that our newly established methods is able to detect E. gingivalis.
Our future goal would be a more comprehensive and professional study in cooperation with a dental clinic, with samples taken by trained personnel, with, amongst others, participants with diagnosed periodontitis and a well worked out questionnaire. However, for more extensive studies very often a smaller study is a prerequisite in order to get the means (or more bluntly spoken the money) and the cooperation from professionals in the field.
With this simple approach we attempted to show that we are capable to detect E. gingivalis with a sensitive and well-working method. By including the information on “gingivitis”, which we changed to self-reported gingivitis, throughout the manuscript, we attempted to show that E. gingivalis might be present very early in periodontal disease, and obtain some very basic indication of the potential pathogenic potential.
The discovery of a new subtype with our method was in our opinion the essential finding of our study, which we hope will facilitate a more comprehensive evaluation of the prevalence of E. gingivalis in the future, since so far PCR protocols employed, did not detect this subtype.
We hope that by rephrasing parts of the manuscript, explaining our approach in more detail and more clearly pointing out the limitations of our study, we can improve the manuscript and put our approach in a better context.
Introduction section
・It is not clear which target diseases are of greatest concern as a result of infection with E.gingivalis; whether the research target is gingivitis or periodontitis. If the authors want to investigate microflora in healthy subjects, it is difficult to understand why gingivitis patients are included. The research objectives should be simple, clear and easily understood by the reader.
Thank you for your comment. We hope by rephrasing the aims of the study, we could set a clearer focus and explain what we wanted to show in this study.
・The authors examined whether E.gingivalis is a factor in the development of gingivitis, but gingivitis without periodontal tissue destruction is not a very serious disease. Gingivitis and periodontitis are also mixed in the text. A clear stating of the disease to be investigated is necessary.
We tried to make this aspect clearer. You are of course right, that gingivitis is not a disease of great concern, but it might be prerequisite for the development of periodontitis. We tried to get a more consistent focus on what we wanted to show and will explain in more detail, what was considered “gingivitis” in this study. As proposed from another reviewer the term “self-reported” gingivitis” is used now and in the method section it is explained in more detail which symptoms were considered for this “self-reported” gingivitis (irritation, bleeding, inflammation, pain).
Material and Methods section
・Basic medical information on the participants is unclear. Specifically, the number of teeth present, oral hygiene status, degree of periodontal disease, dental history, smoking habits, and history of antibiotic use in the past several months. Age and gender alone equals no information.
That is of course a limitation of this study. As mentioned earlier, a detailed questionnaire will be indispensable for a follow-up study. We could try to contact all participants and get some additional information, but this would require very much time and be dependent on the participants compliance and since the sampling was a while ago, some information e.g. antibiotics use would be hard for the participants to remember. Due to the limited number of participants an analysis of these data might also lead to biased interpretation.
We hope that by rephrasing our aims and explaining, why only information on “self-reported” gingivitis was included, will can justify the reduced patient information.
We provided additional information on the sampling process and how self-reported gingivitis was defined in Material and Methods.
・P2L84-90
Shouldn't commercial products be contaminated with E.gingivalis.?
Obviously, they can be contaminated. This was not expected. When we initially used commercially available sterile saline in single use vials (in pharmacytical quality) we were surprised to find out that the single vials of the saline were positive for DNA of E. gingivalis. For Acanthamoeba spp., a free-living amoebae, which we also work on, we know that trances of DNA of these amoebae can even be found in bottled water, which is obviously also a possibility for DNA from E. gingivalis. We tried as thoroughly as possible to eliminate any contaminations by double-autoclaving, sterilizing, and UV-treating all used components and additionally ran several negative controls to minimize the risk of false-positive samples. Since most of the sequences obtained from positive samples were unique, in particular in the ITS regions, we are very confident that no contaminations led to positive results.
・P3L125
The consecutive numbers in the subtitle are wrong. Need to be corrected thereafter.
We are sorry, but obviously your line numbers are different from the numbers in our manuscript, so we are not sure what is meant. We found a mistake for the accession numbers of the EGITS sequences and corrected that.
Discussion section
・P9L267-268
It is unclear why the patients themselves would collect the samples. The method is uncertain and the detailed patient profile is unknown. Are Volunteers Dental Professionals? Patient profiles need to be mentioned in the Methods section.
We hope we explained our approach and the reduced patient information earlier. We have to admit that an unarguable amount of uncertainty remains, when untrained participants collect the samples themselves. That is why we hoped that our initial approach with rinsing the oral cavity would be applicable, since at least it would have been more consistent for all participants. We tried to standardize the procedure as much as possible, gave the participants detailed instructions and observed the sampling process carefully, also aiming to avoid any contaminations.
・P9L269-
If the authors assume that the purpose of the study is to investigate the bacteria that cause periodontal disease, isn't it inconsistent to select a sample of healthy subjects?
We hope that in a future study we will be able to shed some more light on the involvement of E. gingivalis in developing periodontal diseases. This was not the aim in this study and at best our results provide some very basic indication that E. gingivalis might be involved in periodontal diseases, or more accurately are more prevalent in people with minor gingival problems. We hope that we were able to make this clearer and put our results in a more accurate context.
・Throughout the article, there is no mention of bias and no mention of limitation.
We are sorry for that. We hope that we pointed out the limitations more clearly and could explain our approach in more detail.

Reviewer 2 Report
In the manuscript entitled „Pilot study on the prevalence of Entamoeba gingivalis in Austria – detection of a new genetic variant” Köhsler et al. screened 59 individuals for the presence, association with self-reported gingivitis and sequence type of Entamoeba gingivalis.
General comments: Nice study and well written about an under-investigated topic, the oral commensal amoebae
Specific comments.
Abstract: please add the number of voluntary participants and – very important - please never mention “gingivitis” without the adjective "self-reported" (in this case here "with a self-reported history of gingivitis"). check this throughout the entire text.
Introduction.
Line 35: “E” must be italics
Line 44: which ILs?
Line 57: please add “eukaryotic” or “protozoal” to “species” to further differentiate from 16S/23S rRNA/ITS genes in prokaryotic species.
Methods. This is the weak part of the study.
1) Questionnaire: the clinical entity of gingivitis was not measured or clinically assessed but instead was only self-reported by individuals. What was the exact question? Did a 81 old male individual interpret “gingivitis” in the same way then a 21 year old women, probably working in the dental or “prophylaxis” field? Most likely not. Maybe the investigators even asked some of their colleagues within the medical/prophylaxes field (thus educated) and parents (not educated) to participate. Anyway, the definition of "gingivitis" and “GI” in this study here must be taken with reservations; this is the principal reason why this reviewer wants to have “gingivitis" or “GI” always be mentioned with the extension/addendum “self-reported” or “self-diagnosed”. This is also important for Tables 1&2 legends
2) The storage and transportation of samples is not described. Why such a huge 15 ml tube for small interdental brushes and with only 0.2 ml of solution? This could lead to contamination because of dispersion of hand piece surface; better to cut/collect only the tip off brush with a sterile scissor and use a smaller sample and tube. The hint of contaminated water (“occasional detection of E.g. DNA”) is interesting and supports the risks for environmental contamination here. Please discuss.
3) Line 92: how many E.g. DNA sequences were available?
4) Table 1. Please correct “EGPf1rev” (delete P), please add amplicon length for ST2fw/EGF2rev (e.g. repeat EGf2rev below ST2fw and report length in bps)
5) Avoid “used saline” or “utilized saline”; always, as negative control, the best corresponding water/saline is used. This has not to be mentioned explicitly.
6) When listing the samples, use a comma before “and”, e.g. …ALA, BF, and SGL
7) Line 109: Annealing: you mean 45 sec; 45’ would be min, 45’’ would be seconds but to avoid mis-interpretation use “sec” or what is standard for this journal.
8) Line 150: are you sure about the GraphPad Prism version 7, should be 9?!
Results
Figure 1, sample DS; the signal for ST1 is not convincing; better picture? Why is this particular signal so weak?
Discussion:
Line 260: self-reported gingivitis (again check throughout the text).
Line 275: define “periodontitis sites”; again the dental/periodontology expertise could be improved in this team; you mean “inflamed sites” or how were these sites exactly defined in the studies cited (PPD, CAL, BOP, stage, grade etc.)
Final sentence: “Approaches with qPCR”; from the beginning the question arises why no quantitative data were accessed here?
Otherwise, and with the shortcoming of missing dental examination and expertise, a nice study with value for the community.
Author Response
In the manuscript entitled „Pilot study on the prevalence of Entamoeba gingivalis in Austria – detection of a new genetic variant” Köhsler et al. screened 59 individuals for the presence, association with self-reported gingivitis and sequence type of Entamoeba gingivalis.
General comments: Nice study and well written about an under-investigated topic, the oral commensal amoebae
Thank you for your valuable remarks. We hope we considered all of them and could improve the manuscript. Please find changes to the manuscript highlighted in yellow.
Specific comments.
Abstract: please add the number of voluntary participants and – very important - please never mention “gingivitis” without the adjective "self-reported" (in this case here "with a self-reported history of gingivitis"). check this throughout the entire text.
Thank you for your comment. We changed that in the abstract.
Introduction.
Line 35: “E” must be italics
Thank you, we changed that.
Line 44: which ILs?
We added the specific ILs. Thank you.
Line 57: please add “eukaryotic” or “protozoal” to “species” to further differentiate from 16S/23S rRNA/ITS genes in prokaryotic species.
We amended that.
Methods. This is the weak part of the study.
- Questionnaire: the clinical entity of gingivitis was not measured or clinically assessed but instead was only self-reported by individuals. What was the exact question? Did a 81 old male individual interpret “gingivitis” in the same way then a 21 year old women, probably working in the dental or “prophylaxis” field? Most likely not. Maybe the investigators even asked some of their colleagues within the medical/prophylaxes field (thus educated) and parents (not educated) to participate. Anyway, the definition of "gingivitis" and “GI” in this study here must be taken with reservations; this is the principal reason why this reviewer wants to have “gingivitis" or “GI” always be mentioned with the extension/addendum “self-reported” or “self-diagnosed”. This is also important for Tables 1&2 legends
Thank you for your comment. We appreciate the suggestion of “self-reported” gingivitis and agree that under our study conditions this is a more appropriate term. We are aware of the limitations of our study, and should have pointed them out more transparently in the manuscript.
Participants were asked whether they regularly suffer from bleeding (when brushing their teeth), redness, irritation, inflammation or pain of their gums. The answer to these questions is of course strongly depending on the awareness, sensitivity or pain threshold of the participant, and naturally subjective. With this simple approach we wanted to show that we are generally capable to detect E. gingivalis with a sensitive and well-working method. By including the information on “gingivitis” we attempted to show that potentially at a very early stage of periodontal disease the prevalence E. gingivalis might be higher than in healthy persons, providing some basic indication of the pathogenic potential.
In fact, we hope that we might be able to perform a more comprehensive follow-up study, with a more professional approach, more participants, hopefully in cooperation with a dental clinic and trained personnel and a carefully worked out questionnaire. Unfortunately, for a larger study, usually a smaller study is a prerequisite in order to get the means and the cooperation from professionals in the field.
We hope that by providing more information and clearly pointing out the limitations of our study, we can legitimate the weaknesses of our approach.
- The storage and transportation of samples is not described. Why such a huge 15 ml tube for small interdental brushes and with only 0.2 ml of solution? This could lead to contamination because of dispersion of hand piece surface; better to cut/collect only the tip off brush with a sterile scissor and use a smaller sample and tube. The hint of contaminated water (“occasional detection of E.g. DNA”) is interesting and supports the risks for environmental contamination here. Please discuss.
Thanks for the suggestions! Actually, that would have been a safer approach, and will be considered if samples will be taken in a similar way in the future.
Samples were taken in one of our laboratories, where generally no experiments are performed, only the Sequencer is operated. All participants wore sterile gloves and freshly sterilized (autoclaved and UV-treated) material was used. The 15ml tube was chosen for easier handling for the participants. There was no transportation and storage as such. Fresh samples were immediately processed and DNA was isolated or the samples were immediately stored at –20°C.
Since we initially had problems with positive samples and positive controls (these initial samples were of course excluded from the study) we tried to work as thoroughly and careful as possible. The initial idea to use commercially available sterile saline in single-use vials like a mouthwash, was based on the Coronavirus testing system at the time in Austria, where people in their homes would rinse their oral cavity with saline, release the solution into a tube and bring it to collection points for further processing. This method would have been a way to standardize the sample collection to a certain extend and would not have required any professional skills. The sensitivity of this way of sampling was of course not clear to that point. When we detected an unusually high number of positive samples with identical sequences, we tested the saline and found several vials contaminated with E. gingivalis DNA. Therefore, we changed our approach, with additional autoclaving und UV-treating of the saline and additional controls. We felt that we had to mention this in the manuscript. We do not know whether dental saline employed by professionals is additionally treated and free of DNA, which might be the case, however, we felt obliged to spread this information, since the saline we used was a quality product for laboratory use, sterile, but obviously not DNA free.
3) Line 92: how many E.g. DNA sequences were available?
Altogether 32 sequences were available at the time. We used all of them in an alignment for primer design. However, only five of them included the entire 18S rRNA gene, and four of these five, included the ITS regions. These sequences were also used for our phylogenetic analysis (accession numbers are provided in “Phylogeny”. The other available E. gingivalis sequences were shorter and represented different parts of the 18S rRNA gene.
- Table 1. Please correct “EGPf1rev” (delete P), please add amplicon length for ST2fw/EGF2rev (e.g. repeat EGf2rev below ST2fw and report length in bps)
Thank you for noticing. We corrected that and added the amplicon length.
- Avoid “used saline” or “utilized saline”; always, as negative control, the best corresponding water/saline is used. This has not to be mentioned explicitly.
Thank you. We changed that in the manuscript. The reason we mentioned this in more detail, was to demonstrate, that we did not only run the standard no-template-control with the PCR, but also with the saline used for sampling and the components of the DNA isolation kit to rule out any risk of positive results due to potentially contaminated components.
6) When listing the samples, use a comma before “and”, e.g. …ALA, BF, and SGL
Thank you. We corrected that.
- Line 109: Annealing: you mean 45 sec; 45’ would be min, 45’’ would be seconds but to avoid mis-interpretation use “sec” or what is standard for this journal.
Thank you for noticing. We corrected that into s.
8) Line 150: are you sure about the GraphPad Prism version 7, should be 9?!
I checked again and I am afraid we still use version 7.
Results
Figure 1, sample DS; the signal for ST1 is not convincing; better picture? Why is this particular signal so weak?
Depending on the sample some bands were stronger and some were weaker. What we observed was, that people from the “gingivitis” group, more often had stronger bands, which, however, in conventional PCR not necessarily equals more amoebae, but could be a reason for stronger bands. The intensity of the bands did not influence sequencing. We repeated the PCR and replaced the upper part of the figure, now with another ST1 sample (ZM). Unfortunately, it is again a sample with a weaker band compared to the other samples, but stronger than the sample DS. Due to very limited amounts of DNA left, we are afraid that this is the best we can do at short notice. We apologize for that.
Discussion:
Line 260: self-reported gingivitis (again check throughout the text).
We hope we explained what we considered self-reported gingivitis in this study in a more transparent way in the material and methods section and corrected the term throughout the manuscript.
Line 275: define “periodontitis sites”; again the dental/periodontology expertise could be improved in this team; you mean “inflamed sites” or how were these sites exactly defined in the studies cited (PPD, CAL, BOP, stage, grade etc.)
Thank you for the comment. We checked the references. Some authors provided more details (oedema, bleeding), but in general in all these studies “periodontitis sites” were defined as periodontal pockets with a certain depth ranging from 3 to 7mm.
We rephrased the sentence accordingly and added this information.
Final sentence: “Approaches with qPCR”; from the beginning the question arises why no quantitative data were accessed here?
We totally agree with you on that. Unfortunately, there was no particular funding for this study. With our study we hoped to make a first step towards a bigger, more professional project as mentioned earlier, hopefully with less limitations. Establishing a pPCR for the detection of all three subtypes would definitely be a goal of such a project. We hoped that by demonstrating that we are generally capable to detect and identify E. gingivalis, the next steps, applying for funding and establishing a cooperation with a dental clinic, would be more promising.
Otherwise, and with the shortcoming of missing dental examination and expertise, a nice study with value for the community.

Round 2
Reviewer 1 Report
The manuscript has been revised correctly.